# Detectability of Iodine in Mediastinal Lesions on Photon Counting CT: A Phantom Study

**DOI:** 10.3390/diagnostics15060696

**Published:** 2025-03-11

**Authors:** Joric R. Centen, Marcel J. W. Greuter, Mathias Prokop

**Affiliations:** 1Department of Radiology, University Medical Centre Groningen, University of Groningen, 9713 GZ Groningen, The Netherlands; j.r.centen@umcg.nl (J.R.C.); mathias.prokop@radboudumc.nl (M.P.); 2Department of Medical Imaging, Radboud University Nijmegen Medical Centre, 6525 XZ Nijmegen, The Netherlands

**Keywords:** computed tomography, iodine contrasting agent, detectability

## Abstract

**Background/Objectives:** To evaluate the detectability of iodine in mediastinal lesions with photon counting CT (PCCT) compared to conventional CT (CCT) in a phantom study. **Methods:** Mediastinal lesions were simulated by five cylindrical inserts with diameters from 1 to 12 mm within a 10 cm solid water phantom that was placed in the mediastinal area of an anthropomorphic chest phantom with fat ring (QRM-thorax, QRM L-ring, 30 cm × 40 cm cross-section). Inserts were filled with iodine contrast at concentrations of 0.238 to 27.5 mg/mL. A clinical chest protocol at 120 kV on a high-end CCT (Somatom Force, Siemens Healthineers) was compared to the same protocol on a PCCT (Naeotom Alpha, Siemens Healthineers). Images reconstructed with a soft tissue kernel at 1 mm thickness and a 512 matrix served as a reference. For PCCT, we studied the result of reconstructing virtual mono-energetic images (VMIs) at 40, 50, 60 and 70 keV, reducing exposure dose up by 66%, reducing slice thickness to 0.4 and 0.2 mm, and increasing matrix size from 512 to 768 and 1024. Two observers with similar experience independently determined the smallest insert size for which iodine enhancement could still be detected. Consensus was reached when detectability thresholds differed between observers. **Results:** CTDIvol on PCCT and CCT was 3.80 ± 0.12 and 3.60 ± 0.01 mGy, respectively. PCCT was substantially more sensitive than CCT for detection of iodine in small mediastinal lesions: to detect a 3 mm lesion, 11.2 mg/mL iodine was needed with CCT, while only 1.43 mg/mL was required at 40 keV and 50 keV with PCCT. Moreover, dose reduced by 66% resulted in a comparable detection of iodine between PCCT and CCT for all lesions, except 3 mm. Detection increased from 11.2 mg/mL on CCT to 4.54 mg/mL on PCCT. A matrix size of 1024 reduced this detection threshold further, to 0.238 mg/mL at 40 and 50 keV. For 5 mm lesions, this detection threshold of 0.238 mg/mL was already achieved with a 512 matrix. Very small, 1 mm lesions did not profit from PCCT except if reconstructed with a 1024 matrix, which reduced the detection threshold from 27.5 mg/mL to 11.2 mg/mL. Reduced slice thickness decreased iodine detection of 3–12 mm lesions but not for 1 mm lesions. **Conclusions:** Iodine detectability with PCCT is at least equal to CCT for simulated mediastinal lesions of 1–12 mm, even at a dose reduction of 66%. Iodine detectability further profits from virtual monoenergetic images of 40 and 50 keV and increased reconstruction matrix.

## 1. Introduction

Mediastinal lesions encompass a broad spectrum of pathological entities ranging from benign cysts to malignant tumours [1,2]. Accurate characterization of mediastinal lesions is imperative for appropriate clinical management and therapeutic decision-making. Uptake of contrast material serves as an indicator of vascularization and the presence of a solid component within the lesion. High sensitivity to contrast enhancement is therefore important for lesion characterization.

Photon-counting computed tomography (PCCT) represents a recent advancement in CT technology, offering potential radiation dose reduction and enhanced spectral imaging capabilities that provide higher contrast enhancement. Because of direct conversion of x-ray photons into an electrical signal, the photon-counting detector (PCD) elements in PCCT can be made substantially smaller than in conventional CT (CCT), resulting in a higher spatial resolution [3,4,5]. PCDs are capable of detecting each individual x-ray photon and its energy. This method results in the reconstruction of virtual monoenergetic images (VMIs). VMIs in PCCT are synthesized images that simulate the appearance of CT scans as if they were acquired with a single energy level, or keV. This technique leverages the spectral information provided by photon-counting detectors to improve image quality and diagnostic accuracy. Spectral images are of clinical importance due to their improvement in detection and characterization of lesions after use of intra-venous (IV) contrast material [6,7]. PCCT can perform K-edge imaging to distinguish iodine from other substances based on spectral characteristics, offering better quantification and reduced noise compared to traditional CT methods. Moreover, VMIs provide the means for a reduction in the amount of iodine administered to the patient [5,6,8,9,10,11,12].

Detection of contrast enhancement in mediastinal lesions is crucial for their characterization. However, it is not yet known if and to which extent PCCT is superior to CCT and whether dose reduction is possible for this imaging task using this novel technology. Therefore, this phantom study was performed to compare the detectability of iodine in simulated mediastinal lesions with photon counting CT and conventional CT.

## 2. Materials and Methods

An anthropomorphic thorax phantom (QRM-20100, QRM, Freiburg, Germany) with outer dimensions of 200 × 300 mm was used with a fat extension ring (QRM-20108, QRM, Freiburg, Germany) to increase the outer dimensions to 300 × 400 mm (Figure 1). The phantom contained artificial lung lobes, a spine insert, and a soft tissue-equivalent shell. A borehole of 100 mm diameter in the mediastinal region was filled with a cylindrical phantom made of water-equivalent material (Solid Water HE, Sun Nuclear, Melbourne FL, USA) with a mass density of 1.032 g/cm^3^. The mediastinal phantom had five apertures, each with a diameter of 18 mm. Within these apertures, cylindrical inserts of various diameters (1, 3, 5, 8, and 12 mm) were inserted. These inserts were subsequently filled with iodine contrast agent of different concentrations. The study utilized Iomeron 350 (Bracco, Milan, Italy) of 714 mg iodine/mL in mixture with NaCl solution (0.9%) to obtain concentrations of 0.238 ± 0.43, 1.43 ± 0.41, 4.54 ± 0.37, 11.2 ± 0.36 and 27.50 ± 0.62 mg iodine per ml. The standard deviations in the iodine concentrations were derived from the uncertainty in pipetting the volumes of NaCL and Iomeron.

Data were acquired with a clinically used thorax protocol on a Somatom Force CCT and a Naeotom Alpha PPCT (Siemens Healthineers, Forchheim, Germany). The CCT was equipped with an automatic exposure control (CARE kV) which automatically adjusts the tube voltage setting based on patient parameters, such as size and anatomy. The CCT was used as a reference with the following scan parameters used in clinical routine: 120 kV (based on CARE kV), 50 quality refence mAs, Br40 kernel and 1 mm slices. For the PCCT, the following four parameters were varied: (1) VMI levels of 40–70 keV, (2) dose level from 34 to 100% of the reference dose, (3) matrix size of 768 and 1024 pixels, and (4) slice thickness in high resolution 0.4 mm and ultra-high resolution 0.2 mm (Table 1).

The detectability of the inserts, categorized by diameter and iodine concentration, for all acquisition and reconstruction settings was determined by two independent observers who were instructed to scroll through the slices. Both observers were masters biomedical engineering students with 5 and 3 years of experience in radiology. Inter- and intra-observer variability were calculated with Cohen’s Kappa statistic per concentration. To be able to construct contrast-detail diagrams, consensus was reached in case of discrepancies. If the detectability of an insert was scored different by both observers, they reached consensus by discussing their observation.

Mean noise levels were determined for CCT and PCCT by measuring and averaging the standard deviation of CT numbers in 4 regions of 2.5 cm^2^ within the mediastinal phantom around the central iodine insert. In addition, the mean CT values of the just detectable inserts were measured on CCT and PCCT as a function of VMI, dose level, slice thickness, and matrix size.

## 3. Results

The noise measured on CCT was 23.7 ± 1.2 HU, while the PCCT exhibited noise levels of 24.8 ± 2.6, 19.6 ± 1.5, 14.8 ± 0.9 and 12.6 ± 1.1 HU for VMI 40, 50, 60 and 70 keV, respectively. CTDIvol of 3.80 ± 0.12 mGy and 3.60 ± 0.01 mGy was measured at 100% dose for CCT and PCCT, respectively. The results of the CT value measurement of the just detectable iodine inserts are shown in the Appendix A. The CT value on CCT increased from 37 to 302 HU for the 1.43 to 27.5 mg/mL insert, respectively. On PCCT, the CT values increased with lower VMI and were approximately stable under a variation in dose, slice thickness and matrix sizes. The inter-observer Cohen’s Kappa values with 95% confidence intervals were 0.756 ± 0.083, 0.859 ± 0.018, 0.926 ± 0.022, 0.966 ± 0.024 and 1.00 ± 0.017 for the 0.238 ± 0.43, 1.43 ± 0.41, 4.54 ± 0.37, 11.2 ± 0.36 and 27.5 ± 0.62 mg/mL concentrations, respectively.

The effect of VMI on lesion detectability is demonstrated for the example of 3 mm lesions in Figure 2.

Differences between PCCT and CCT were most pronounced for lesions of 3 mm and 5 mm diameter (Figure 3A). Detectability relative to CCT improved for VMI of 60 and 70 keV but was highest for VMI of 40 and 50 keV (Figure 3A), decreasing the detection threshold by a factor of 7.8 for 3 mm lesions and a factor of 19.1 for 5 mm lesions. Because it had the highest detection performance and a lower noise level than 40 keV VMI, the 50 keV VMI was further used for the analyses of the effect of dose reduction, matrix size and reconstructed slice thickness. While dose reduction decreased lesion detectability with PCCT compared to CCT at full dose, dose reduction to 42% and 52% still yielded improved lesion detectability compared to CCT, and dose reduction to 34% yielded at least equal detectability compared to CCT (Figure 3B). Decreasing reconstructed slice thickness decreased lesion detectability (Figure 3C), but even at 0.2 mm slice thickness, lesion detectability with PCCT was at least equal to that with CCT.

For larger lesions of 8 mm and 12 mm diameter, detectability was, in general, better than for smaller lesions, but differences in detectability with PCCT were less pronounced compared to CCT. We found identical results for CCT and PCCT VMI of 60 and 70 keV. Only at 40 and 50 keV could the lowest contrast concentrations be detected with PCCT, which made it superior to CCT (Figure 3A). This better detectability vanished with dose reduction but, again, detectability with PCCT at even 34% of the reference dose yielded equal results to full-dose CCT (Figure 3B). Thinner slices with PCCT led to at least equal detectability to CCT, even for the thinnest sections of 0.2 mm (Figure 3C).

The smallest lesion (1 mm) required very high iodine concentrations (27.5 mg/mL) to be detected, independent of whether CCT or PCCT was used. It was not improved by virtual monoenergetic reconstructions (Figure 3A) but also did not suffer with dose reduction in PCCT (Figure 3B) or thinner reconstructed sections (Figure 3C). Only by increasing the matrix size from 512 to 1024 could the detection threshold be reduced to 27.5 mg/mL on PCCT with 50 keV reconstructions (Figure 3D).

This effect of improved detectability with larger matrix size was also seen for 3 mm lesions, in which the detectability could be further reduced to the lowest concentration of 0.238 mg/mL. An intermediate matrix size of 768 did not affect lesion detectability.

## 4. Discussion

From this phantom study on the detectability of mediastinal lesions in PCCT, we can conclude that iodine detectability at the standard setting of 60 keV is at least equal to CCT for simulated mediastinal lesions of 1–12 mm and that iodine detectability further increases with virtual monoenergetic images of 40 and 50 keV for all lesions ≥ 3 mm. Moreover, chest CT with PCCT at 66% dose reduction yields an iodine detectability in mediastinal lesions that is at least equal to that of CCT for all lesions.

Optimal detectability of iodine in mediastinal lesions was found by VMI reconstructions at 40 and 50 keV. VMI reconstructions have also shown added benefit in spatial resolution, noise texture and contrast-to-noise ratio (CNR) on a photon-counting dual-layer CT system by Ozguner and colleagues [13]. They illustrated that the noise texture was overall the highest at VMIs of 40 keV and decreased at higher monoenergetic reconstruction levels. This was also found by Greffier et al. who showed on a prototype PCCT that noise was reduced, noise texture was improved, and spatial resolution and detectability on VMIs for all low-keV levels was better compared to CCT [14]. Bette et al. also concluded that PCCT VMIs in oncologic patients demonstrated reduced image noise compared to CCT and improved the conspicuity of hypervascularized liver metastasis at low-keV settings [15]. In the current study, as a trade-off between iodine detectability and noise, we used a VMI level of 50 keV in our analysis for the influence of dose, slice thickness and matrix size for the detectability of iodine in mediastinal lesions in photon-counting CT.

Dose reduction with PCCT relative to CCT was also found justified for coronary CT angiography [3]. The results showed an optimized detectability index for low (40) and high (350) HU values on PCCT versus CCT. Moreover, the detectability index was more than two times higher for these high and low HU values, even though CCT used 19% more dose. For the detection of pulmonary embolism, Pannenbecker et al. found that a reduction of 51% in effective radiation dose was achievable in PPCT in comparison to CCT without loss of subjective image quality and diagnostic confidence [16]. For the detection of liver lesions, which are larger than coronary lesions or pulmonary emboli, Racine et al. were only able to show a general trend towards dose reduction with PCCT [17]. All these studies showed that a decrease in radiation dose is attainable when comparing PCCT with CCT. In the current study, dose reductions up to 66% on PCCT were feasible without decreasing the detectability of ≥3 mm lesions. These results were realized due to VMI reconstructions near the k-edge of iodine, the increased accuracy of PCCT compared to CCT and the superior spatial resolution of the PCCT system [4,18].

Kawata et al. studied the impact of slice thickness on the detection of hypervascular hepatocellular carcinomas [19]. They concluded that while detection of lesions increased with thinner slices, noise also increased. Our study showed that PCCT allowed for using thinner sections (down to 0.2 mm) without losing the detectability of lesions relative to 1 mm slices with CCT. For superior detectability of lesions compared to CCT, however, PCCT had to be reconstructed at 1 mm slices. The loss of lesion detectability with thinner slices can be explained by the increase in quantum mottle [20].

The use of larger matrix sizes has also shown to maintain spatial resolution and improve image quality of lung diseases [21]. The current study demonstrated increased visibility of 1 mm and 3 mm iodine inserts by increasing the matrix size to 1024. However, no effect was observed for larger inserts, nor for 3 mm inserts at a matrix size of 768.

This study had limitations. First, the CCT and PCCT used in this study utilized different reconstruction methods. Although both were iterative reconstruction methods, CCT used ADMIRE strength 3 while PCCT used QIR strength 2. However, according to vendor information, the ADMIRE strength 3 reconstruction on CCT is closest to the QIR strength 2 reconstruction on PCCT. A similar approach has been used by McCollough et al. [22]. Second, this is a phantom study on the detectability of mediastinal lesions. The conclusions on optimal VMI level, dose reduction, iodine reduction, slice thickness and matrix size for detection must be taken with care since more research has to be performed in order to verify these results in an in vivo setting.

While we have explored the effect of iodine detectability in simulated mediastinal lesions, we have not simultaneously looked at spatial resolution, which may also be influenced by low-keV VMI. Current image reconstruction with iterative or deep learning techniques tends to focus on noise reduction but does so at the expense of spatial resolution. Spatial resolution becomes dose- and contrast-dependent [23,24,25]. The effect is least pronounced for high-contrast objects (like lesions with high iodine content reconstructed at 50 keV) but is most pronounced for low-contrast objects close to the detection threshold. While our work supports superior lesion detectability with 50 keV VMI with PCCT, it allows for no conclusion about other factors affecting image quality.

The fact that we used (almost) identical exposure settings for CCT and PCCT is fair from a physics point of view and has been used in most previous studies that compare PCCT to CCT [15,26,27]. However, it may not represent clinical practice: while contrast-enhanced chest imaging on CCT usually employs lower tube voltage settings of 80 to 100 kVp, depending on patient size, clinical PCCT data are usually acquired with 120 kVp to allow for better energy separation in order to optimize the image quality of virtual monoenergetic images. This has two consequences: lower kVp at CCT will require less exposure dose than 120 kVp, and lower kVp will increase enhancement of iodine, thus potentially improving iodine detectability. While our study confirmed the superiority of PCCT with 50 keV VMI at 120 kVp for detection of objects with low iodine enhancement compared to CCT obtained with identical parameters, conclusions may not be directly translated to settings in which low-kVp imaging is used for CCT. This effect is the consequence of different parameter optimizations for CCT and PCCT, and how to tackle this has not yet been fully solved. Correlation with clinical studies will be necessary to support superior performance of PCCT in a clinical setting. We therefore plan to extend our measurements with lower tube voltages such as 100 and even 80 kVp to assess whether lower doses and superior detectability of iodinated mediastinal lesions are also feasible with PCCT.

## 5. Conclusions

In summary, our study shows improved detection of iodine enhancement in simulated small mediastinal lesions using 50 keV VMI with PCCT in comparison to CCT obtained at similar exposure settings. Substantial dose reductions or substantially thinner slice thicknesses are possible with PCCT compared to CCT without compromising iodine detectability. Increasing the reconstruction matrix size to 1024 improves iodine detectability in the smallest structures of 1–3 mm diameter.

## Figures and Tables

**Figure 1 diagnostics-15-00696-f001:**
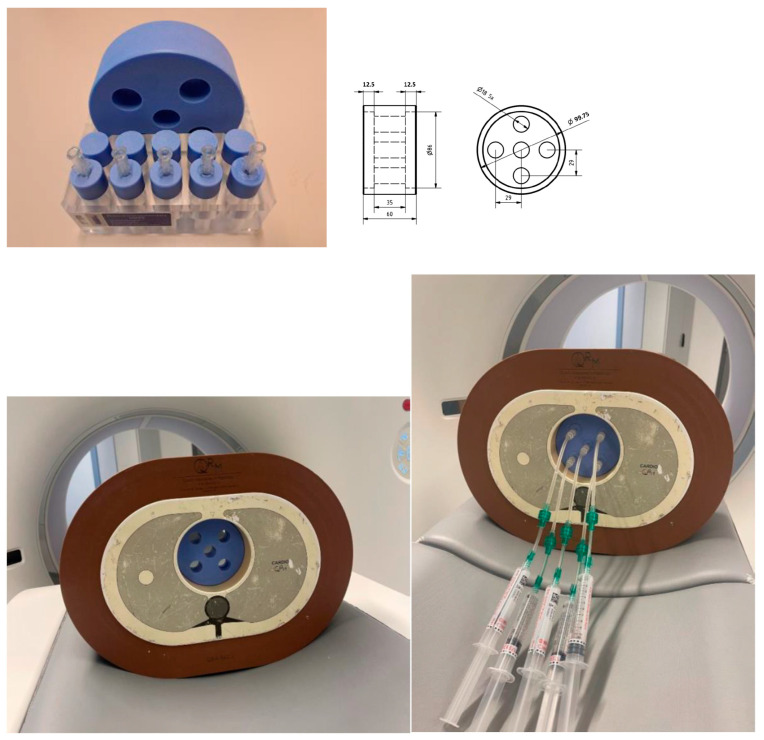
Anthropomorphic thorax phantom with extension ring, and in the centre the home-made cylindrical insert with 5 holes of 18 mm × 35 mm, which can be filled with cylindrical lesions of 1, 3, 5, 8 and 12 mm in diameter.

**Figure 2 diagnostics-15-00696-f002:**
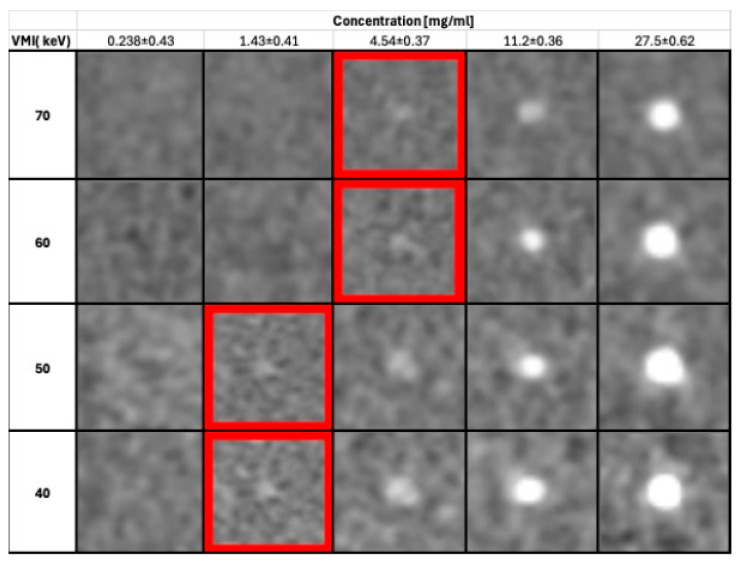
Detectability of 3 mm lesions per virtual monochromatic image (VMI) reconstruction as a function of iodine concentration. Highlighted in red are the minimal concentrations needed to detect the lesion at VMI levels of 40–70 keV (WW/WL: 400/40).

**Figure 3 diagnostics-15-00696-f003:**
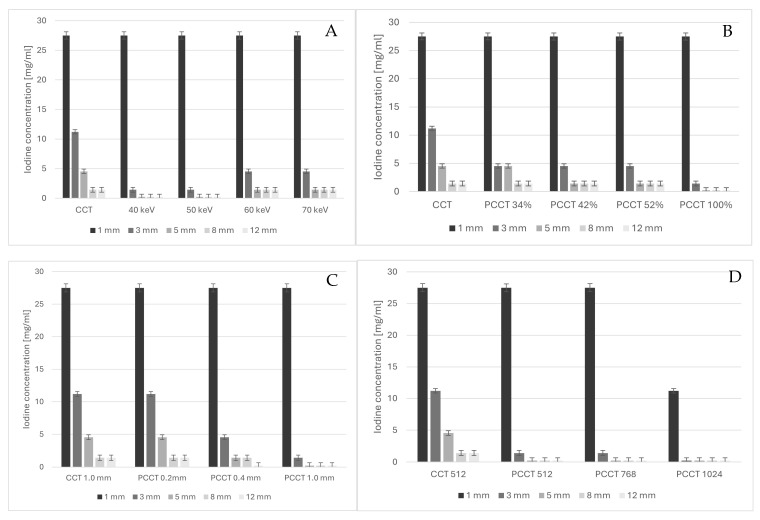
Detectability (defined as minimal detectable iodine concentration per lesion diameter) as a function of virtual monochromatic image reconstruction at 40, 50, 60 and 70 keV and 100% dose on conventional CT (CCT) and photon-counting CT (PCCT) (**A**), as a function of dose on PCCT at 34, 42, and 52% of reference PCCT and CT dose (**B**), as a function of slices thicknesses of 0.2, 0.4 and 1.0 mm (**C**), and as a function of matrix sizes of 512, 768 and 1024 (**D**).

**Table 1 diagnostics-15-00696-t001:** Acquisition and reconstruction parameters on conventional and photon-counting CT.

	Conventional CT	Photon-Counting CT
Acquisition mode	Spiral	Spiral/quantum plus
Tube voltage [kV]	120	120
qref mAs/IQ level	50	34/42/52/100%
Automatic Exposure Control	On	On
Collimation [mm]	96 × 0.6	120 × 0.2
Field of view [mm]	250	250
Rotation time [s]	0.5	0.5
Pitch	0.6	0.6
Slice thickness [mm]	1.0	1.0/0.4/0.2
Increment [mm]	1.0	1.0
Reconstruction kernel	Br40	Br40
Matrix size [pixels]	512	512/768/1024
Reconstruction method	ADMIRE 3	QIR 2
Virtual Monochromatic Image [keV]	n/a	40/50/60/70

n/a = not applicable; ADMIRE 3 = Advanced Modelled Iterative Reconstruction strength 3; QIR 2 = Quantum Iterative Reconstruction strength 2.

## Data Availability

The dataset supporting the conclusions of this article are available upon request to the corresponding author.

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
