# Peer review of "Detectability of Iodine in Mediastinal Lesions on Photon Counting CT: A Phantom Study"

_diagnostics, 2025, doi:10.3390/diagnostics15060696_

Round 1

Reviewer 1 Report

Comments and Suggestions for Authors

This is nice and fair study aiming to evaluate the detectability of iodine in mediastinal lesions with photon counting CT compared to conventional CT. Similar studies has been already performed before (not only for CCT, but also for DECT). As expected (and already shown before) substantial dose reduction is possible using photon counting compared to conventional CT without compromising iodine detectability.

Main concern in this study is that authors claim to compare use of CT and PCCT in specific clinical situation but use inadequate protocol for conventional CT (120 kV). This is noted as limitation by authors. Explanation given by the authors that 120 kV in CCT was used as ‘fair from physics point of view, and other studies are performed in similar way’ should be better justified. Especially since PCCT image acquisition is completely different from CCT and similar voltage does not mean that similar information will be obtained. Additionally, please explain why ADMIRE 3 and QIR 2 reconstruction methods are used (how these matches each other)? It would be more useful and interesting to see how PCCT perform in comparison with CT using clinical protocol tailored for this clinical situation in both units. Especially because authors state in the Intro:

‘However, it is not yet known if and to which extent PCCT is superior to CCT, and whether dose reduction is possible for this imaging task using this novel technology.’

After this study we still do not know since CCT protocol performed in this study is not used clinically.

Additionally,

Abbreviations are not used properly sometimes. Please check whole text: e.g. CCT is used without explanation in intro, PCD (is it photon counting detector?), IV (intravenous?). In Meth and Mat section’ Neothom Alpha PPCT’ is used? This is PCCT probably.

Results are written with a lot of discussion within. This makes Results section unclear. Please put only results in this section and Figures should follow immediately after referenced in the text.

Figure 2 is missing?

Please check the text. Sometimes it is not clear or there are missing words? E.g.

‘They illustrated that the noise texture and was overall the highest at..’

‘…VMIs for all low-keV levels was better compared to [14].’

Please rewrite Discussion para starting with:

‘Dose reduction with PCCT relative to CCT was found justified for coronary CT angiography[3].’

It is not clear why this sentence is here till the whole paragraph was red.

Reviewer 2 Report

Comments and Suggestions for Authors

In their manuscript "Detectability of iodine in mediastinal lesions on photon counting CT: a phantom study", Centen et al. investigated the advantages of PCCT in the discrimination of small lesions in a chest phantom. As the authors point out, with the increasing use of PCCT, the discrimination of small iodine-enhancing lesions may be of interest in a clinical setting.
The present manuscript is well written and conclusive. However, several issues need to be addressed before publication can be considered:

General aspects:

  • In the study, the discreteness of the lesions appears to be based solely on the subjective observation of two investigators. There are probably no significance levels given in this context either. The study design is also limited by the lack of direct measurement of the HU values in the inserts. This should be done before publication and would clearly substantiate the results of the otherwise merely subjective study.
  • Since the conventional scanner used is a dual-energy scanner, why were the results not compared to other kV settings in conventional CT?

Abstract:

Fine, no comments.

Introduction:

p2 ll49-58: Please add the explanation for the presumably better iodine detection (iodine k-edge)

p2 ll54-55: “This method results in the reconstruction of virtual monoenergetic images (VMI) also known as spectral imaging.“ Misleading, please rephrase and explain in a little more detail.

Materials and Methods:
- Was the iodine measurable within the inserts (e.g., single pixel measurement tools)? Measurement of HU values within the inserts would be of interest.
- Were sham inserts used in the study?

p2 l80: Please explain the rationale for using 1 mm slice thickness, why not 0.6 mm? Different reconstructed slice thicknesses from raw data would be of interest. This is highlighted in the results section, but Figure 5 does not seem to match the figure legend.

p3 l85: "by two independent observers" Radiologists? Trained in CT diagnosis?

p3 l88: "Consensus was reached in case of discrepancies." Please elaborate.

Results:

p4 l108: typo: …concentrations, respectively.

Figure 2: not visible in the reviewer's version.

Figure 4/5: figures look exactly the same. It seems figure 5 was not uploaded.

All figures: No significance values are shown.

Discussion:

p l174: typo: “Who”

p7 ll208ff: Limitations are highlighted in detail. However, as noted above, some limitations are still missing.

Round 2

Reviewer 1 Report

Comments and Suggestions for Authors

Authors replied or changed the manuscript in satisfactory manner.